# A Highly Expressed Antennae Odorant-Binding Protein Involved in Recognition of Herbivore-Induced Plant Volatiles in *Dastarcus helophoroides*

**DOI:** 10.3390/ijms24043464

**Published:** 2023-02-09

**Authors:** Shan-Cheng Yi, Yu-Hang Wu, Rui-Nan Yang, Dong-Zhen Li, Hazem Abdelnabby, Man-Qun Wang

**Affiliations:** 1Hubei Insect Resources Utilization and Sustainable Pest Management Key Laboratory, College of Plant Science and Technology, Huazhong Agricultural University, Wuhan 430070, China; 2Key Laboratory of Forest Protection of National Forestry and Grassland Administration, Ecology and Nature Conservation Institute, Chinese Academy of Forestry, Beijing 100091, China; 3Department of Plant Protection, Faculty of Agriculture, Benha University, Banha 13736, Qalyubia, Egypt

**Keywords:** herbivore-induced plant volatiles, odorant-binding proteins, *Dastarcus helophoroides*, binding affinities, behaviorally attractive substances

## Abstract

Natural enemies such as parasitoids and parasites depend on sensitive olfactory to search for their specific hosts. Herbivore-induced plant volatiles (HIPVs) are vital components in providing host information for many natural enemies of herbivores. However, the olfactory-related proteins involved in the recognition of HIPVs are rarely reported. In this study, we established an exhaustive tissue and developmental expression profile of odorant-binding proteins (OBPs) from *Dastarcus helophoroides*, an essential natural enemy in the forestry ecosystem. Twenty DhelOBPs displayed various expression patterns in different organs and adult physiological states, suggesting a potential involvement in olfactory perception. In silico AlphaFold2-based modeling and molecular docking showed similar binding energies between six DhelOBPs (DhelOBP4, 5, 6, 14, 18, and 20) and HIPVs from *Pinus massoniana*. While in vitro fluorescence competitive binding assays showed only recombinant DhelOBP4, the most highly expressed in the antennae of emerging adults could bind to HIPVs with high binding affinities. RNAi-mediated behavioral assays indicated that DhelOBP4 was an essential functional protein for *D. helophoroides* adults recognizing two behaviorally attractive substances: *p*-cymene and γ-terpinene. Further binding conformation analyses revealed that Phe 54, Val 56, and Phe 71 might be the key binding sites for DhelOBP4 interacting with HIPVs. In conclusion, our results provide an essential molecular basis for the olfactory perception of *D. helophoroides* and reliable evidence for recognizing the HIPVs of natural enemies from insect OBPs’ perspective.

## 1. Introduction

The insect olfactory system is essential for maintaining an insect’s life. Many important physiological behaviors, such as host-seeking, mating, and oviposition, primarily depend on sensitive olfactory organs, mainly the antennae [1,2,3]. In these processes, odorants in the environment, such as pheromones and plant volatiles, are considered vital semiochemicals [4,5]. Among them, herbivore-induced plant volatiles (HIPVs) are essential plant secondary metabolite that provide information on the host’s location to natural enemies [6,7,8,9].

The accurate identification of environmental odorants relies on a susceptible olfactory system. Several proteins, such as odorant-binding proteins (OBPs), chemosensory proteins (CSPs), odorant degrading enzymes (ODEs), odorant receptors (ORs), and sensory neuron membrane proteins (SNMPs) in insects’ antennae are involved in the perception processes of the external environment [1,10,11,12]. Among these proteins, OBPs are carrier proteins that transport hydrophobic odorant molecules across hydrophilic lymph to ORs in sensillar neurons [13,14]. As its function as the first filter of olfactory information, combined with RNA interference (RNAi) experiments, OBPs have been investigated in many studies, some of which have confirmed their olfactory function [15,16,17].

Previous studies have shown that the specific expression patterns of OBPs are typically related to their physiological functions [18,19,20,21]. Most OBPs with actual olfactory functions are highly or expressed explicitly in antennae [22,23,24,25,26]. This correlation between function and expression patterns is also present in insect larvae. Two GOBPs from *Plutella xylostella* highly expressed in larval antennae show strong binding affinities with the sex pheromone (Z)-11-hexadecenal, and they suggested that this specific expression is related to the tendency of larvae to food sources containing sex pheromones [27].

The research on protein 3D structures also provides new insights to explore the functions of insect OBPs. A recent study of the protein structure of *Drosophila melanogaster* OBP28a augmented its original predicted function of buffering rapid odor variation in the odorant environment [28]. It demonstrated its essential role in detecting the specific compound β-ionone [29]. Based on OBPs’ primary structures, combined with in silico modeling, molecular docking, in vitro binding assays, and reverse chemical ecology were proposed and successfully implemented in several cases [23,30,31,32,33]. For example, based on the binding affinities of *Aenasius bambawalei* AbamOBP1 evaluated via fluorescence competitive binding assays (FCBAs) and fluorescence quenching assays, 2, 4, 4-trimethyl-2-pentene and 1-octen-3-one were screened. Further behavioral assays showed that these two substances elicited attractive behavioral responses in *A. bambawalei*.

*Dastarcus helophoroides* (Fairmaire) (Coleoptera: Bothrideridae) is an effective biological control agent against *Monochamus alternatus*, which is the vector of the pinewood nematode for *Bursaphelenchus xylophulus* (Steiner et Buhrer) Nickle (Nematoda: Aphelenchoididae) [34,35]. Although previous studies showed that DhelOBP21 was involved in the recognition process of β-pinene [24], we are still in a fog regarding the host localization process of these critical natural enemies. Here, we characterized detailed and comprehensive tissue and developmental expression profiles using a quantitative real-time polymerase chain-reaction (qRT-PCR). The potential binding affinities of six DhelOBPs (DhelOBP4, 5, 6, 14, 18, and 20) with different expression characteristics and HIPVs from *Pinus massoniana* were evaluated using a reverse chemical ecology-based method, followed by in vitro fluorescence competitive binding assays. The physiological function of the corresponding DhelOBP was verified using a Y-tube olfactometer combined with RNAi experiments. Finally, the binding conformations of functional DhelOBP with active behavioral substances were analyzed to investigate their potential key binding sites.

## 2. Results

### 2.1. Analyses of HIPVs of P. massoniana Lamb

GC-MS analyses showed more than 35 peaks in both treatments of healthy and infested wood sections of *P. massoniana* Lamb (Figure 1). In total, 26 substances were identified, and terpenoids were the main components of volatiles from both treatments, among which monoterpenes were the most abundant (Appendix A). A total of 11 of 26 newly appearing or significantly increased substances were considered as HIPVs (Table 1); all the identified HIPVs are monoterpenes.

### 2.2. Odorant-Binding Proteins in D. helophoroides

For identification and classification of DhelOBPs, multiple amino acid sequence alignments were generated using the MAFFT program (https://www.ebi.ac.uk/Tools/msa/mafft/, accessed on 11 March 2021) using 23 DhelOBPs (Appendix A) and 50 OBPs from *Tribolium castaneum* [36] (Appendix A). Based on the above alignment results, we provided more concise alignment results in Figure 2. Most DhelOBPs belong to the classic OBP subfamily and share a conserved cysteine pattern (C1X_24-29_C2X_3_C3X_22-43_C4X_8-10_C5X_8_C6) comparable to *T. castaneum* and other insects. However, one of the classic OBP, DhelOBP22, is similar to TcOBP8B, which has an extra cysteine of 16 AA before C1 and another additional cysteine of 8 AA after C6. Four DhelOBPs (DhelOBP5, DhelOBP6, DhelOBP11, and DhelOBP22) belong to the Minus-C OBP subfamily and also show a conserved cysteine pattern, with only four cysteines lacking C2 and C5 (C1X_18-30_C3X_38-39_C4X_16-17_C6). The one Plus-C OBP, DhelOBP10, has an expanded N and C terminus containing six additional cysteines and a typical proline residue following C6. DhelOBP2 is a partial coding sequence without the C-terminal; we speculated that it might be a Minus-C OBP because it lacks C2 based on the current alignment. DhelOBP1 and DhelOBP23 were identified as complete coding sequences but was not consistent with the conserved sequence motif of any OBP family. As a result, we removed these three DhelOBPs (DhelOBP1, DhelOBP2, and DhelOBP23) in the subsequent studies.

### 2.3. Spatio-Temporal Expression Patterns of DhelOBPs

To investigate the transcript abundance of 20 *DhelOBPs* in female and male adults, we measured the expression patterns in multiple organs (including antennae, head, thorax, abdomen, reproductive organ, leg, and wing) at different adult physiological states (emerging, virgin, and mated). Firstly, all amplification efficiencies for these twenty *DhelOBPs* and three reference genes (*RPS*, *GAPDH*, *α-Tubulin*) were tested in order to be directly used in the equation from Vandesompele et al. [37] (Appendix A). The results showed that 20 *DhelOBPs* shared similar tissue and developmental expression patterns between female and male adults (Figure 3); however, the expression levels of different DhelOBPs in the antennae varied greatly (Figure 4). As shown in Figure 3, eight of these *DhelOBPs* (*DhelOBP4*, *9*, *13*, *14*, *15*, *16*, *18*, and *19*) were specifically expressed in adults’ antennae. The remaining 12 *DhelOBPs* were also expressed in other organs. *DhelOBP6*, *DhelOBP11*, and *DhelOBP21* were almost expressed in all organs. In particular, *DhelOBP3* and *DhelOBP20* were mainly expressed in the head. Intriguingly, the expression levels of *DhelOBPs* in antennae also differed considerably. *DhelOBP4* was the most highly expressed *OBP* in the antennae of emerging adults (Figure 4A,B), while *DhelOBP9* was the most highly expressed *OBP* in the antennae of virgin and mated adults (Figure 4C–F). In addition to *DhelOBP4* and *DhelOBP9*, several *DhelOBPs* such as *DhelOBP5*, *DhelOBP13*, *DhelOBP14*, *DhelOBP15*, *DhelOBP17*, *DhelOBP18*, and *DhelOBP21*, showed consistently high expression levels in antennae at different adult physiological states. Correspondingly, *DhelOBP3*, *DhelOBP7*, *DhelOBP12*, and *DhelOBP20* showed extremely low expression levels compared to others at all tested adult physiological states. Furthermore, *DhelOBPs* in antennae also showed a considerable variation in expression with adult physiological states. Thirteen *DhelOBPs* (*DhelOBP3*, *7*, *8*, *9*, *11*, *12*, *13*, *14*, *15*, *16*, *18*, *19*, and *22*) were upregulated with the adult physiological state, and conversely, four *DhelOBPs* (*DhelOBP4*, *5*, *6*, and *10*) were downregulated (Appendix A).

### 2.4. AlphaFold2-Based Modeling and Molecular Docking of DhelOBPs

We selected three DhelOBPs (DhelOBP4, DhelOBP14, and DhelOBP18) that are specifically and highly expressed in antennae (DhelOBP4, DhelOBP14, and DhelOBP18), two DhelOBPs (DhelOBP5 and DhelOBP6) which are expressed in multiple organs, and one DhelOBP (DhelOBP20) which mainly expressed in the head as target proteins to conduct modeling and molecular docking. The AlphaFold2-Based Colab server (https://colab.research.google.com/github/deepmind/alphafold/blob/main/notebooks/AlphaFold.ipynb, accessed on 13 May 2022) was used to model the 3D structure of our target proteins. Several programs including pLDDT, ERRAT, VERIFY 3D, and PROCHECK were applied to evaluate the qualities of the 3D structures. All parameters indicated that the 3D modeling of the above six DhelOBPs were reliable (Appendix A). The protein structures of DhelOBP4, 5, 6, 14, 18, and 20 contained six typical α helices and an internal cavity, which defines a hydrophobic binding cavity (Figure 5). Then, all HIPVs were chosen as ligands to conduct molecular docking with target DhelOBPs, including α-fenchene, *p*-cymene, γ-terpinene, terpinolene, fenchone, camphor, pinocamphone, isopinocamphone, terpinen-4-ol, α-terpineol, and (-)-verbenone. Similar binding energy values ranging from −5.4 to −7.4 kcal mol^−1^ were observed between all tested DhelOBPs and ligands (Table 2). As a result, eight available HIPVs (except α-fenchene, pinocamphone, and isopinocamphone) were subjected to subsequent fluorescence competitive binding assays (FCBAs).

### 2.5. Binding Characteristic of Recombinant DhelOBPs

Recombinant proteins of DhelOBP4, 5, 6, 14, 18, and 20 were obtained using a prokaryotic expression system. The produced proteins were successfully expressed as soluble forms in *Escherichia coli*. Purified recombinant proteins on SDS-PAGE are presented in Figure 6A. A single protein band for each recombinant protein was seen on the gel with the predicted protein molecular weight (Appendix A). Purified recombinant mature proteins were obtained by cleaving the His-tag off using the TEV protease.

*N*-phenyl-1-naphthylamine (1-NPN) was chosen as a fluorescent probe to measure the binding affinities between DhelOBPs and HIPVs. Six recombinant DhelOBPs (DhelOBP4, 5, 6, 14, 18, and 20) exhibited a regular saturation binding curve to 1-NPN and a linear Scatchard plot (Figure 6B,C). The dissociation constants (*K_d_*) of DhelOBP4, 5, 6, 14, 18, and 20 bound to 1-NPN at pH 7.4 were 22.42, 40.98, 120.48, 17.39, 1.23, and 5.13 μM, respectively. As shown in Figure 6D–I and Table 3, only DhelOBP4 showed general binding affinities with HIPVs at pH 7.4 (*K_i_* < 30 μM), and the binding affinities decreased significantly at pH 5.0. The binding affinities of the other four DhelOBPs, DhelOBP5, 14, 18, and 20, were too weak (IC50 > 50 μM) to calculate accurate *K_i_* values. For DhelOBP6, all tested HIPVs increased the fluorescence intensity, making the results impossible to be identified. More detailed information on FCBAs is shown in Appendix A.

### 2.6. Behavioral Response of D. helophoroides to HIPVs with the Best Binding Affinities

Three HIPVs with the best binding affinities to DhelOBP4, α-terpineol (10.4 ± 0.1 μM), γ-terpinene (11.0 ± 0.3 μM), and *p*-cymene (13.0 ± 0.9 μM) were selected for the following Y-tube olfactometer experiments. The results revealed that *D. helophoroides* were significantly attracted by γ-terpinene (*p* = 0.0237 and 0.0477 for female and male adults, respectively) and *p*-cymene (*p* = 0.0477 and 0.0477 for female and male adults, respectively) (Figure 7A,B). However, α-terpineol (*p* = 0.5716 and 0.3961 for female and male adults, respectively) did not elicit a behavioral response from *D. helophoroides*.

To determine the effect of decreasing the expression level of *DhelOBP4* on *D. helophoroides* behavior choice, the adults were injected with *dsDhelOBP4* and *dsGFP*, then examined using a qRT-PCR. As expected, a significant reduction in the transcript of DhelOBP4 occurred in the *dsDhelOBP4* treatment, with 85.22% and 94.70% decreases compared with the *dsGFP* treatment on the 6th day post-injection for female and male adults, respectively (Figure 7C,D). Then, we conducted Y-tube olfactometer experiments to determine whether *DhelOBP4* deficit in *D. helophoroides* influences their behavioral responses to γ-terpinene and *p*-cymene. The results revealed that *D. helophoroides* with *DhelOBP4* deficiency showed a significant loss of tendency to γ-terpinene and *p*-cymene compared to the dsGFP treatment (Figure 7E–H). Notably, after silencing *DhelOBP4*, *p*-cymene and γ-terpinene elicited a significant repellent response from female adults (Figure 7E,F), while they did not elicit a behavioral response from male adults (Figure 7G,H).

### 2.7. Binding Conformation of DhelOBP4 with HIPVs

The post-RNAi behavioral assays indicated that DhelOBP4 might play a crucial role for *D. helophoroides* in detecting HIPVs components. To better understand the binding mechanisms of DhelOBP4, we further analyzed the key binding sites and essential amino acid residues of DhelOBP4 binding with HIPVs. The residues of DhelOBP4 interacting with HIPVs are presented in Figure 8 and Table 4. The hydrophobic residues most commonly interacting between DhelOBP4 and HIPVs included Leu 11, Met 50, Phe 54, Val 56, Phe 71, and Phe 126. Four hydrophilic residues, including His 51, Tyr 90, Cys 93, and Gly 113, were exposed to the binding cavity, possibly contributing to forming hydrogen bonds with ligands. Two active behavioral compounds, γ-terpinene, and *p*-cymene, shared the exact same ligand interaction mechanisms with DhelOBP4. For α-terpineol, which showed no behavioral effect on *D. helophoroides*, an additional hydrogen bond with residue His 51 was observed. The function of the key residues warrants further study using mutant proteins.

## 3. Discussion

Chemical communication is vital to the tri-trophic interactions among host plants, herbivores, and their natural enemies [4,38,39]. Previously, we have identified 23 putative OBP transcripts in *D. helophoroides* antennae using a transcriptomic approach [40]. However, in this study, we excluded two DhelOBPs (DhelOBP1 and DhelOBP23), which did not show a conserved sequence motif of any OBP subfamily (Figure 2). Meanwhile, we also speculated DhelOBP2 was a putative Minus-C OBP as its C-terminal lost partial coding sequence and showed conserved C1, C3, and C4. Of course, a 3′ RACE experiment is required to confirm this speculation in the future. Of the remaining 20 DhelOBPs, fifteen of them were Classic OBP, four of them were Minus-C OBP, and one of them was Plus-C OBP. Compared to other coleopteran insects, *D. helophoroides* showed a relatively average repertoire of OBPs (21 OBPs), which is similar to *Holotrichia parallela* (25 OBPs) [41], *Harmonia axyridis* (19 OBPs) [42], *Colaphellus bowringi* (26 OBPs) [43], and *Tenebrio molitor* (19 OBPs) [44]. Meanwhile, more OBPs were found in *Anoplophora chinensis* (46 OBPs) [45], *Semanotus bifasciatus* (32 OBPs) [46], *M. alternatus* (29 OBPs) [40], and *T. castaneum* (50 OBPs) [36]. The variation in the number of chemosensory genes could be linked to an adaptation to living habitats and other environmental factors. The average repertoire of OBPs is just one aspect of chemosensory genes in *D. helophoroides*; more aspects of the comparative research, including CSPs, ORs, IRs, and SNMPs, are required to explore the relationship between chemosensory genes and their living habitat.

Tissue-specific and developmental expression patterns of OBPs are typically associated with their potential physiological functions. We performed a thorough and comprehensive tissue and developmental expression profiling study. We found that *DhelOBP4*, *5*, *9*, *13*, *14*, *15*, *17*, *18*, and *21* were predominantly expressed in adults’ antennae with continuously high transcript levels (Figure 3 and Figure 4), implying their potential roles in olfactory perception. Conversely, four *DhelOBPs* (*DhelOBP3*, *7*, *12*, and *20*) shared deficient transcript levels in antennae at all tested adult physiological states. However, *DhelOBP3* and *DhelOBP12* were predominantly expressed in the head. The actual expression levels need to be further verified at the protein level using a Western blot. As reported by Yang et al. [47], they found DhelOBP10 was highly expressed at the protein level in *D. helophoroides* unmated adults; however, its transcript expression level was significantly lower than many *DhelOBPs* in this study (Figure 4A–D). They also found that *M. alternatus* and *D. helophoroides* relied on the evolutionarily conserved OBPs—MaltOBP24 and DhelOBP10, respectively, to recognize (+)-fenchone. We also analyzed the expression of the OBP-encoding genes at different adult physiological states (emerging, virgin, mated). Other genes exhibited considerable variation at different adult physiological states (Appendix A). For example, *DhelOBP9* and *DhelOBP14* were upregulated (by more than fivefold) in virgin and mated adults compared with emerging adults, suggesting that they might be linked to reproductive behavior or host-seeking for oviposition.

Six DhelOBPs (DhelOBP4, 5, 6, 14, 18, and 20) were evaluated using a reverse chemical ecology method to determine whether they could be involved in the recognition processes of HIPVs. Although they showed similar binding energies calculated via modeling and molecular docking (Table 2), the in vitro FCBAs revealed that only DhelOBP4 showed broad binding abilities (*K_i_* < 30 μM) with all tested HIPVs (Table 3). However, the FCBAs might also present some inaccurate results. A recent study indicated that the calculated affinities of FCBAs could be quite different when using different fluorescent reporters. In this study, the binding affinities of DhelOBP4, 5, 6, 14, 18, and 20 assessed via only one fluorescent reporter, 1-NPN, might also be imprecise. More fluorescent probes such as the 1-aminoanthracene and even more in vitro binding assay techniques are required to comprehensively determine these proteins’ binding properties. Nevertheless, the strong binding abilities of DhelOBP4 with HIPVs imply that it might be a potential olfactory-functional protein.

Based on DhelOBP4′s binding affinities, three HIPVs with the best binding affinities (α-terpineol, γ-terpinene, and *p*-cymene) were evaluated using behavioral assays. Y-tube olfactometer experiments indicated that γ-terpinene and *p*-cymene showed significant attractional effects on female and male *D. helophoroides* adults compared with the control (Figure 7A,B). Further RNAi-mediated behavioral experiments demonstrated the olfactory function of DhelOBP4 during the detection of γ-terpinene and *p*-cymene by *D. helophoroides*, and both females and males lost their tendency responses to these two substances. The recognition of odorant substances by insects requires the corresponding OBPs to transport them across the lymph, where the ORs ultimately completes the signal transduction and elicits the behavioral responses of insects [10,48]. In this context, we hypothesized that in the absence of DhelOBP4, the key ORs that elicit the behavioral responses could not complete the signal transduction, resulting in adults losing their tendency to γ-terpinene and *p*-cymene. However, γ-terpinene and *p*-cymene could still elicit a significant repellent response in female adults without DhelOBP4 (Figure 7E,F). It seemed to indicate that those female adults were more sensitive to HIPVs to the extent that they could still exhibit behavioral responses to both substances even with the loss of DhelOBP4. From an evolutionary point of view, more sensitive olfactory systems might help female adults locate their hosts more accurately and find a better place for mating or oviposition. Although the molecular mechanisms underlying the recognition of these two substances in the absence of DhelOBP4 by females were not clear, we could still identify DhelOBP4 as an important protein for the recognition of γ-terpinene and *p*-cymene by *D. helophoroides*.

Binding specificity has always been attributed to ligand structures and OBP 3D structures during their interactions. Similar ligands structures were observed in two attractive substances, γ-Terpinene and *p*-cymene, which mainly rely on hydrophobic force to bind with DhelOBP4 (Figure 8). The additional hydrogen bond formed between DhelOBP4 and α-terpineol might be responsible for the stronger binding affinity compared to γ-Terpinene and *p*-cymene. Furthermore, γ-terpinene and *p*-cymene shared exactly the same residues when the bond to DhelOBP4, among which Phe 54, Val 56, and Phe 71 were the most commonly interacted residues of DhelOBP4, interacted with all the tested HIPVs. The functions of these critical sites can be verified via site-directed mutagenesis in the later stages.

A deep understanding of insect olfactory mechanisms will help us to regulate insect behavior more effectively for pest control. Although our study identified two attractive substances and the key OBP involved in their recognition, the above queries remain unexplained, revealing more about an complexity of an insect’s olfaction. In the future, more ideas should be developed to explain the olfactory sensory mechanisms of insects and their potential applications for pest control.

## 4. Methods and Materials 

### 4.1. Insect Culturing and Pine Woods Collection

*D. helophoroides* pupae were provided by the Research Institute of Forest Ecology, Environment, and Protection, Chinese Academy of Forestry. Adults were fed on the dried meat floss of *M. alternatus* larvae and pupae at 25 ± 1 °C. The healthy and infested *Pinus massoniana* Lamb wood sections were cut from the non-infected and infected areas of *M. alternatus* in Yichang, Hubei Province, China.

### 4.2. Collection and Analysis of P. massoniana HIPVs

After the fresh wood sections were cut down from the tree, they were immediately cut into 0.4 m sections to collect volatiles using a closed-loop dynamic headspace sampling system, as previously described by Sun et al. [49]. One milliliter of n-hexane was used to elute volatiles from traps. Plant volatiles were collected for 24 h at a 12:12 h light/dark cycle, and seven biological replicates were conducted for each treatment (healthy or infested by *M. alternatus*). After the volatiles collection was completed, all wood sections were cut into pieces to confirm whether they were healthy or infested by *M. alternatus* larvae. The collected volatiles were sent to gas chromatography-mass spectrometry (GC-MS) on a QP-2010 GC-MS instrument (Shimadzu, Japan) equipped with an HP-5 MS fused-silica column (30 m × 0.25 mm × 0.25 μm) (Agilent Technologies Inc., Santa Clara, CA, USA). Helium (1 mL/min) was used as the carrier gas, and the procedure was the same as previously reported by Sun et al. [50].

### 4.3. Classification of DhelOBPs

The nucleotide and amino acid sequences of 23 DhelOBPs were downloaded from the GenBank database (https://www.ncbi.nlm.nih.gov/genbank/, accessed on 13 February 2020) and GenPept database (https://www.ncbi.nlm.nih.gov/protein, accessed on 13 February 2020), respectively (Appendix A). Signal peptide sequences and molecular weights were predicted using the SignalP-5.0 server (http://www.cbs.dtu.dk/services/SignalP/, accessed on 13 February 2020) and ExPASy Compute pI/Mw tool (https://web.expasy.org/compute_pi/, accessed on 13 February 2020), respectively (Appendix A). To increase the accuracy of the sequence alignment and facilitate the classification of DhelOBPs, 50 mature OBPs from *T. castaneum* [36] and 23 mature DhelOBPs were aligned using the MAFFT program (https://www.ebi.ac.uk/Tools/msa/mafft/, accessed on 13 February 2020). Then ESPrint 3.0 (https://espript.ibcp.fr/ESPript/cgi-bin/ESPript.cgi, accessed on 13 February 2020) was used to visualize the results of multiple sequence alignments.

### 4.4. Tissue Sampling, RNA Extraction, and cDNA Synthesis

Emerging, virgin, and mated *D. helophoroides* adults were distinguished between males and females for the following tissue sampling. All dissecting tools were treated for RNase inactivation. Each whole insect was dissected into seven parts: antennae (An), head (H), thorax (T), abdomen (A), reproductive organ (R), leg (L), and wings (W). Then the dissected tissue was immediately placed in a RNase-free tube and stored in liquid nitrogen.

Total RNA was extracted from the above-mentioned preserved tissue using TRIzol reagent (Invitrogen, Carlsbad, CA, USA) and isolated following the manufacturer’s instructions. The purity of the total RNA was determined via agarose gel electrophoresis, and the concentration was measured using a spectrophotometer (Eppendorf Bio Photometer Plus, Hamburg, Germany). cDNA was prepared from 0.5 μg of total RNA via a reverse transcription, using a PrimeScript II 1st Strand cDNA Synthesis Kit (TaKaRa Bio, Otsu, Japan) following the manufacturer’s instructions.

### 4.5. Spatio-Temporal Expression Analyses

The expression level of verified *DhelOBPs* genes was analyzed using a reverse transcription-mediated qRT-PCR. A qRT-PCR was performed on a LightCycler^®^ 96 System using the Hieff^®^ qPCR SYBR Green Master Mix (No Rox) (YEASEN, Shanghai, China). Conditions were the following: 95 °C 5 min, 40 × (95 °C 10 s, 60 °C 25 s), and completed with a final cycle for the post-amplification dissolution curve. Every reaction was systematically run in triplicate with three independent biological replicates. All *DhelOBPs* gene-specific primers were designed using Primer-BLAST (https://www.ncbi.nlm.nih.gov/tools/primer-blast/index.cgi?LINK_LOC=BlastHome, accessed on 22 April 2020) (Appendix A). Normalized expression and relative fold change were calculated based on a model by Vandesompele et al. [37] for normalization against several reference genes when the efficiencies of target and reference genes are not similar. In this study, ribosomal protein (*RPS*), 3-glyceraldehyde phosphate dehydrogenase (*GAPDH*), and the *α*-*Tubulin* gene were chosen as reference genes and were used in each cDNA sample to correct the relative expression levels of *DhelOBPs*. The amplification efficiency of each pair of primers was determined by generating standard curves with mixed cDNA samples (Appendix A). The following equation from Vandesompele et al. [37] was applied for the calculations (*E*: primer efficiency, *Cq*: threshold cycle, *Ref*: reference gene, *Tar*: target gene):$$ratio=\frac{\sqrt[{n}]{{\left(1+E_{Ref1}\right)}^{Cq(Ref1)}{\times (1+E_{Ref2})}^{Cq(Ref2)}\times \cdots }}{{\left(1+E_{Tar}\right)}^{Cq(Tar)}}$$

### 4.6. AlphaFold2-Based Modeling and Molecular Docking

The 3D structures of target DhelOBPs were predicted with the AlphaFold Colab at https://colab.research.google.com/github/deepmind/alphafold/blob/main/notebooks/AlphaFold.ipynb, accessed on 13 May 2022. This Colab does not use templates. The qualities of the models were evaluated using the AlphaFold metric called the predicted local distance difference test (pLDDT) on a scale from 0 to 100, as well as ERRAT, VERIFY 3D, and PROCHECK programs (https://saves.mbi.ucla.edu/, accessed on 15 May 2022). Molecular docking was conducted to determine the mode of binding of ligands using Autodock Vina version 1.2.0 (Scripps Research Institute, La Jolla, CA, USA). The AutoDockTools version 1.5.7 software was employed to generate the docking input files. The search grids (binding pockets) were identified via the DeepSite program (https://www.playmolecule.com/deepsite/, accessed on 15 May 2022) and GetBox Plugin of PyMOL version 2.5.3 software (Schrodinger, New York, NY, USA). To increase the docking accuracy, the value of exhaustiveness was set to 20. For Vina docking, the default parameters were used as described in the Autodock Vina manual unless specified. The top ranked pose judged via the Vina docking score was subject to visual analysis using MOE 2019.0102 (Chemical Computing Group Inc., Montreal, QC, Canada).

### 4.7. Subcloning, Expression, and Purification of Target DhelOBPs

The primers without signal peptides and containing homology arms of target *DhelOBPs* were designed to homologous recombine with pET32b (Appendix A). Recombinant plasmids were transformed into DH5α *Escherichia coli.* (chemically competent cells) (TransGen, Beijing, China) were grown on lysogeny broth (LB) solid medium containing ampicillin (50 μg/mL). Colonies were selected and then sequenced.

For the expression of recombinant proteins, each pET32b vector containing the target sequence was used for transformation into BL21 (DE3) *Escherichia coli* (chemically competent cells) (TransGen, Beijing, China). The positive clone verified using a PCR was inoculated in a 5 mL LB medium with ampicillin (50 μg/mL) with shaking at 200 rpm at 37 °C. After 8 to 10 h, the culture was diluted to 1 L LB medium and grown to an optical density of 0.8 to 1.0 at a wavelength of 600 nm. The protein expression was induced by adding isopropyl-beta-D-thiogalactopyranoside to a final concentration of 0.5 mM, followed by culturing for 14 h at 200 rpm at 16 °C. Then the culture was harvested by centrifugation and sonicated. The protein was purified using a combination of chromatographic steps on anion-exchange resins and His-tag affinity chromatography. All purification steps were monitored using SDS-PAGE.

### 4.8. Fluorescence Competitive Binding Assays

Purified proteins were dissolved in 30 mM Tris-HCl buffer (pH 7.4 and 5.0), and emission fluorescence spectra were recorded on an RF-5301PC fluorescence spectrophotometer (Shimadzu, Kyoto, Japan) at room temperature in a right-angle configuration with a 1-cm light path quartz cuvette. To measure the affinity of the fluorescent probe 1-NPN with the target proteins, a two mM solution of protein in 30 mM Tris-HCL (pH 7.4 and 5.0) was titrated with aliquots of 1 mM 1-NPN to final concentrations ranging from 0 to the approaching saturation. The probe 1-NPN was excited at 337 nm, and the emission spectra were recorded between 350 and 500 nm. The binding affinities of other ligands were measured using 1-NPN (2 μM) as the fluorescent probe with a stoichiometry of 1:1 (protein: probe), with the final concentration of each ligand ranging from 0 to 20 μM. All ligands, including 1-NPN were purchased from Sigma-Aldrich (St Louis, MO, USA) and TCI Shanghai (Shanghai, China) and dissolved in chromatogram class methanol (Aladdin, Shanghai, China).

The dissociation constants (*K_d_*) of DhelOBPs with 1-NPN were calculated from Scatchard plots of the binding data using PRISM 8.0 software (GraphPad, La Jolla, CA, USA). Binding affinities (*K_i_*) of DhelOBPs for each ligand were calculated based on the IC50 value (the initial fluorescence level of the competitor concentration reduced by half) using the equation: $K_{i}=\left[\mathrm{IC}50\right]/(1+\frac{\left[1-\mathrm{NPN}\right]}{K_{1-\mathrm{NPN}}})$, where [1-NPN] is the free concentration of 1-NPN, and *K*_1-NPN_ is the dissociation constant of the complex OBP/1-NPN. The analysis supposes that there is only one protein binding site, which is 100% active, and that the binding is reversible and at an equilibrium.

### 4.9. Y-Tube Olfactometer Experiments

Y-tube olfactometer experiments were conducted to evaluate the behavioral response to HIPVs with the best binding affinities in darkness at approximately 25 ± 1 °C. The central tube was 15 cm long, and the two branching arms, each linked to one vial, were 12 cm long with a 75° angle between them. Twenty microliters of each chemical that had been diluted 1000 times by volume were added to a 1 × 1 cm filter paper in one of the vials, and 20 μL of liquid paraffin were dropped on the same size filter paper in the second vial as the blank control. An effective choice was recorded when adults crawled at least 5 cm into the arm within 1 min and stayed 30 s at least. After every five individuals, the two arms were exchanged to avoid position effects. After testing 10 individuals, the Y-tube olfactometer was cleaned with absolute ethanol and oven-dried at 150 °C for 2 h. Fifty adult *D. helophoroides* were analyzed in each treated group. The results were analyzed using the chi-square test.

### 4.10. Synthesis and Microinjection of dsRNA

dsRNA was generated via an in vitro transcription with completed coding sequence templates amplified via a PCR using the primers DhelOBP4 RNAi-F (5′–TAATACGACTCACTATAGGGATGAAATCGTATTTGGTGTTTTTATGT–3′) and DhelOBP4 RNAi-R (5′–TAATACGACTCACTATAGGGTTACATGAAAGGATCTTCAATGCC–3′), which contained T7 RNA polymerase sequence (indicated by the underline) added to their 5′ ends. dsRNA synthesis was performed using a T7 RiboMAXTM Express RNAi System (Promega, Madison, WI, USA). The dsRNA concentration was examined using a spectrophotometer at 260 nm and was then dissolved in diethyl pyrocarbonate-treated water to a final concentration of 5 μg/μL and was stored at −80 °C until use.

50 male and 50 female adults of *D. helophoroides* were chosen on day 1–4 after eclosion. The two treated groups were: dsGFP-injected and dsRNA-injected. dsGFP (250 ng) or dsRNA (250 ng) was respectively injected into the thorax internode of the adults with a microinjector (World Precision Instruments Inc., Sarasota, FL, USA) under a microscope. The RNAi efficiency was detected using a qRT-PCR for seven consecutive days. Following behavioral response was evaluated using the Y-tube olfactometer.

## Figures and Tables

**Figure 1 ijms-24-03464-f001:**
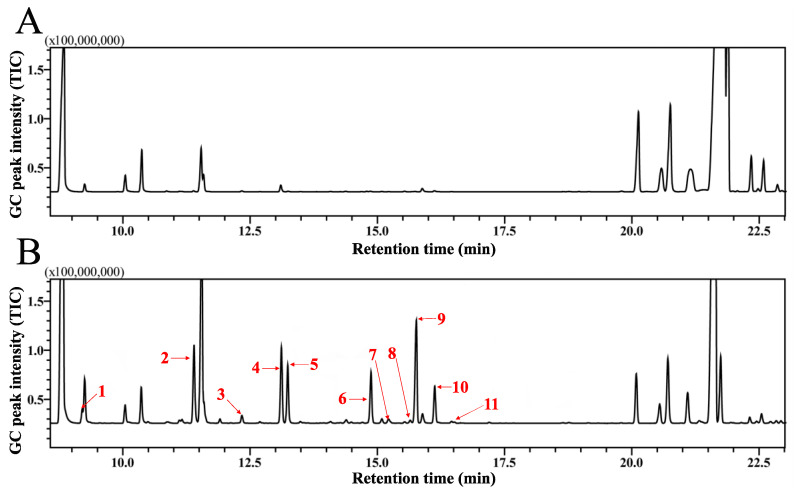
GC–MS chromatograms of *Pinus massoniana* volatiles. (**A**) The GC–MS chromatograms of healthy *P. massoniana* wood sections. (**B**) The GC–MS chromatograms of infested *P. massoniana* wood sections caused by *M. alternatusC*; the HIPVs were marked by red arrows with the number corresponding to Table 1.

**Figure 2 ijms-24-03464-f002:**
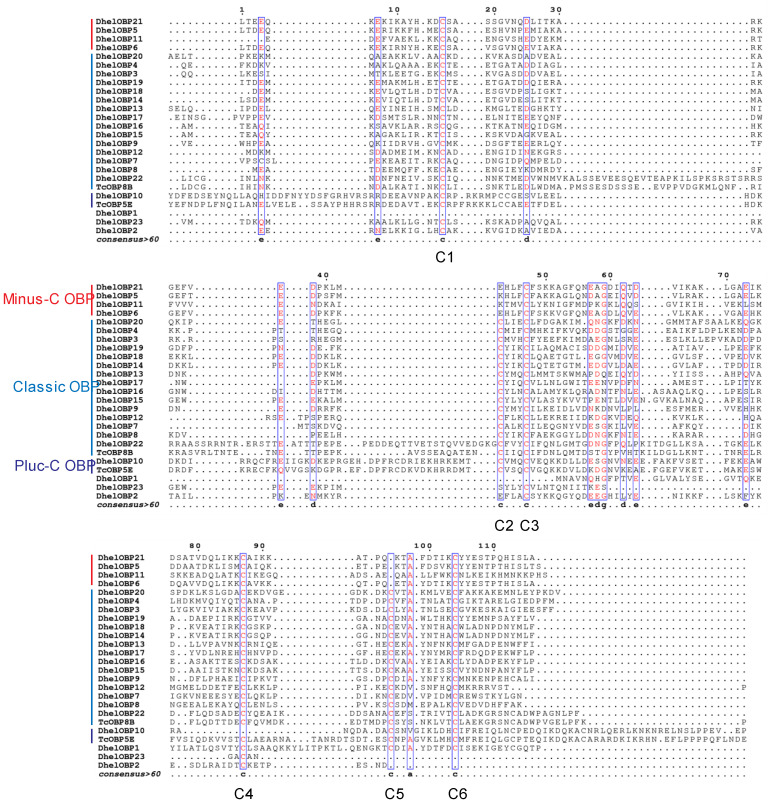
Multiple sequence alignment of OBPs from *Dastarcus helophoroides* and *Tribolium castaneum*. The complete alignment results were presented in Appendix A. Conserved cystine was annotated according to the classic OBP subfamily.

**Figure 3 ijms-24-03464-f003:**
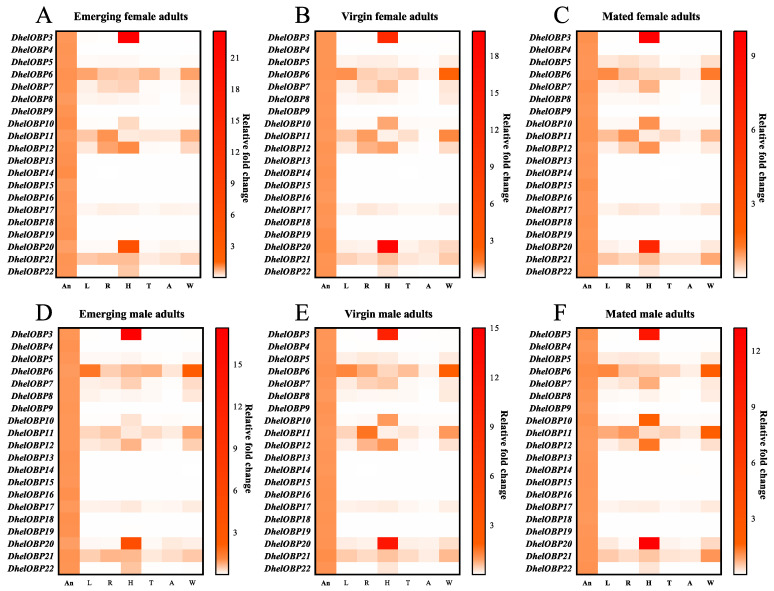
Heatmap of tissue expression patterns of DhelOBPs at different adult physiological states for female and male adults. Transcript levels of 20 DhelOBPs were normalized by *RPS*, *GAPDH*, and *α-Tubulin* genes in seven organs: antennae (An), head (H), thorax (T), abdomen (A), reproductive organ (R), leg (L), and wings (W). (**A**–**F**) are the expression levels of *DhelOBPs* in different organs at different physiological states (emerging, virgin, and mated).

**Figure 4 ijms-24-03464-f004:**
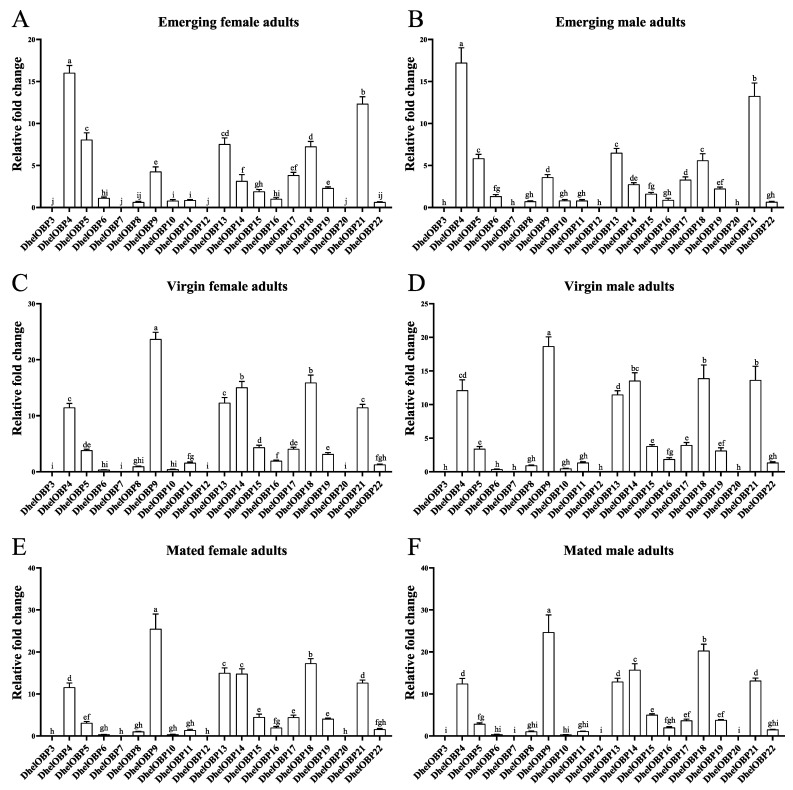
Comparison of transcript levels of different DhelOBPs in antennae for female and male adults. (**A–F**) are the expression levels of *DhelOBPs* in adults antennae at different physiological states (emerging, virgin, and mated). The results were analyzed using one-way ANOVA with Tukey’s HSD post hoc test. Same letters, no significant difference; Different letters, significant difference.

**Figure 5 ijms-24-03464-f005:**
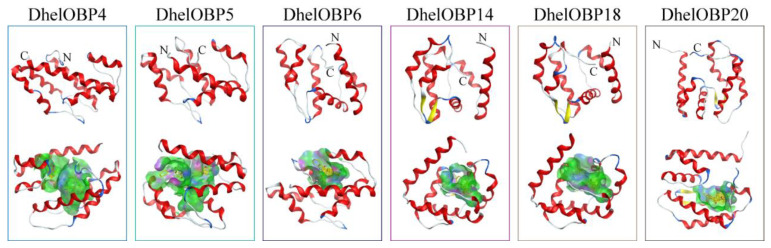
3D structure models and binding cavities of target DhelOBPs. The binding cavities were formed by docking with 1-NPN and all tested HIPVs.

**Figure 6 ijms-24-03464-f006:**
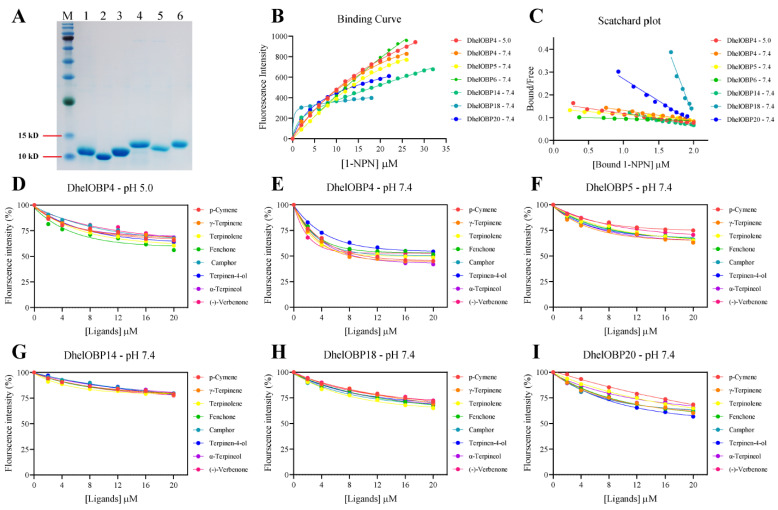
(**A**) SDS-PAGE analysis of target DhelOBPs. M, molecular marker; 1, DhelOBP4; 2, DhelOBP5; 3, DhelOBP6; 4, DhelOBP14; 5, DhelOBP18; 6, DhelOBP20. (**B**) Binding curve for 1-NPN to target DhelOBPs. (**C**) Scatchard plot of target DhelOBPs. (**D**) Binding curve for HIPVs to DhelOBP4 at pH 5.0. (**E**–**I**) Binding curve for HIPVs to DhelOBP4, DhelOBP5, DhelOBP14, DhelOBP18 and DhelOBP20 at pH 7.4, respectively.

**Figure 7 ijms-24-03464-f007:**
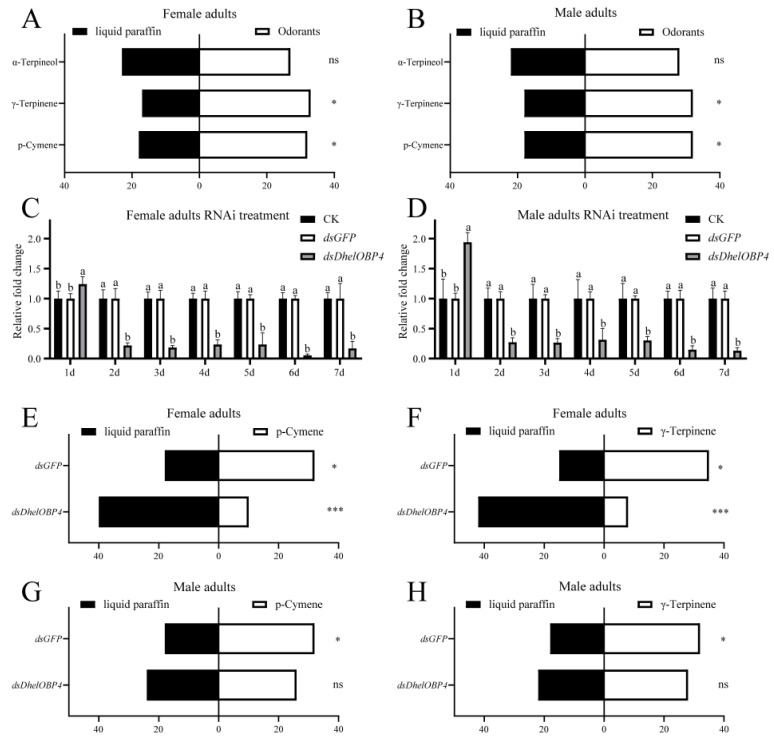
Y-tube olfactometer bioassays and RNAi-mediated bioassays. Same letters, no significant difference; Different letters, significant difference. ns indicates no significant difference; *, 0.01 < *p* < 0.05; ***, *p* < 0.001. (**A**,**B**) Behavioral response of *D. helophoroides* to α-terpineol, γ-terpinene, and *p*-cymene. α-Terpineol showed no behavioral effects on female (*p* = 0.5716, chi-square = 0.3200) and male (*p* = 0.3961, chi-square = 0.7200) adults. γ-Terpinene and *p*-cymene showed attractive effects on female (*p* = 0.0237, 0.0477 and chi-square = 5.1200, 3.9200, respectively) and male (*p* = 0.0477, 0.0477 and chi-square = 3.9200, 3.9200, respectively) adults. (**C**,**D**) Transcript levels of DhelOBP4 post-RNAi and the results were analyzed using one-way ANOVA with Tukey’s HSD post hoc test. (**E**–**H**) Behavioral response of *D. helophoroides* to *p*-cymene and γ-terpinene post-RNAi treatment. *p*-Cymene and γ-terpinene showed identically attractive effects on females (*p* = 0.0477, 0.0477 and chi-square = 3.9200, 8, respectively) and males (*p* = 0.0477, 0.0477 and chi-square = 3.9200, 3.9200, respectively) adults after injection of *dsGFP*. After injection of *dsDhelOBP4*, *p*-cymene and γ-terpinene showed significantly avoidant effects on female adults (*p* < 0.0001, <0.0001 and chi-square = 23.1200, 18.0000, respectively), and no behavioral effects on male adults (*p* = 0.3961, 0.7773 and chi-square = 0.7200, 0.0800, respectively). *p*-values of Y-tube olfactometer bioassays were determined using the chi-square test.

**Figure 8 ijms-24-03464-f008:**
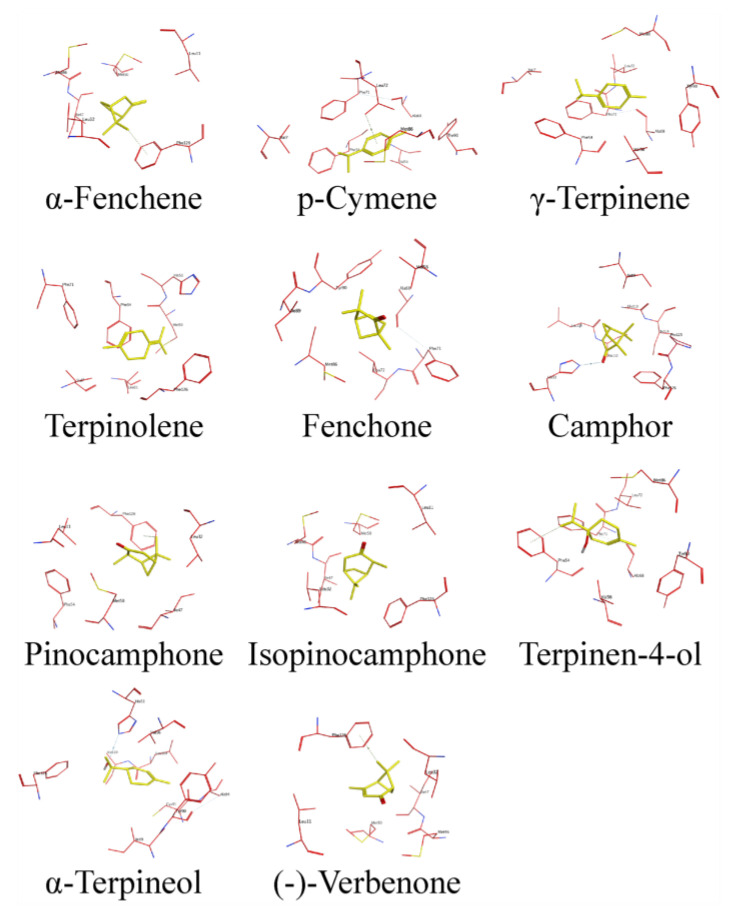
Docking conformation of HIPVs into the putative binding pocket of DhelOBP4.

**Table 1 ijms-24-03464-t001:** Information of HIPVs from infested *Pinus massoniana* Lamb.

Number	Substance	Retention Time (min)	PubChem CID	CAS Number	2-D Structures
1	α-Fenchene	9.260	28930	471-84-1	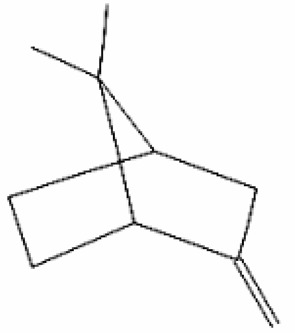
*2*	*p*-Cymene	11.454	7463	99-87-6	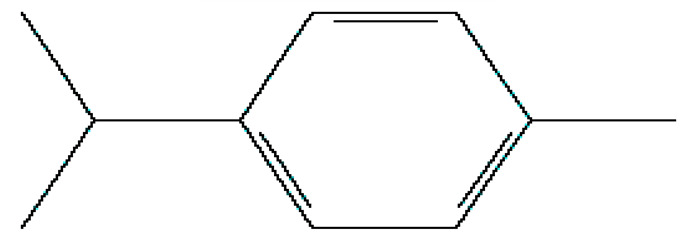
3	γ-Terpinene	12.389	7461	99-85-4	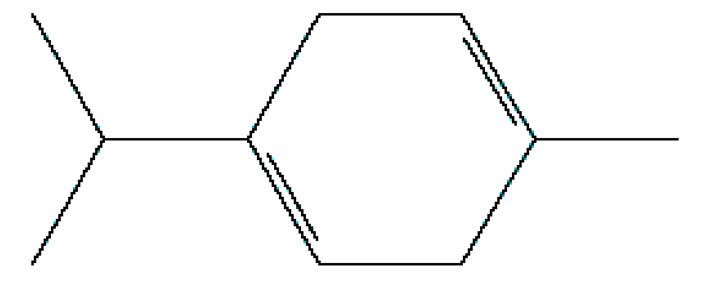
4	Terpinolene	13.178	11463	586-62-9	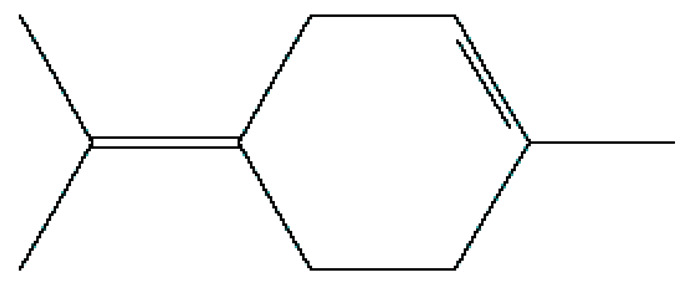
5	Fenchone	13.312	14525	7787-20-4	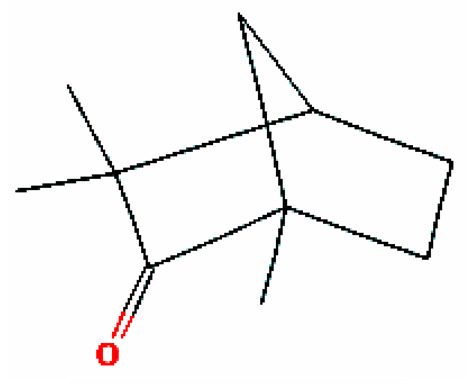
6	Camphor	14.939	2537	76-22-2	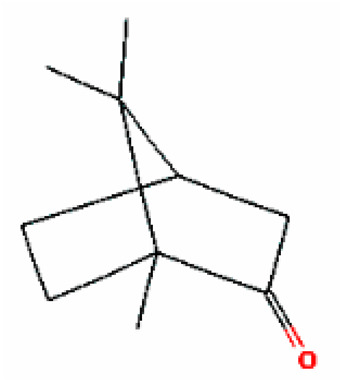
7	Pinocamphone	15.272	6427105	547-60-4	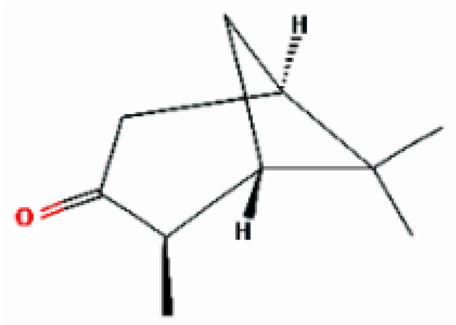
8	Isopinocamphone	15.707	84532	14575-93-0	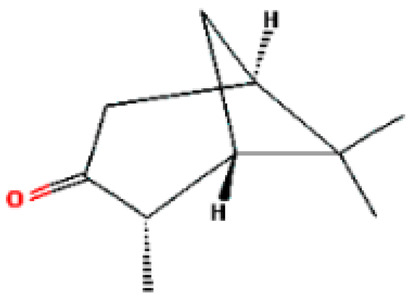
9	Terpinen-4-ol	15.820	11230	562-74-3	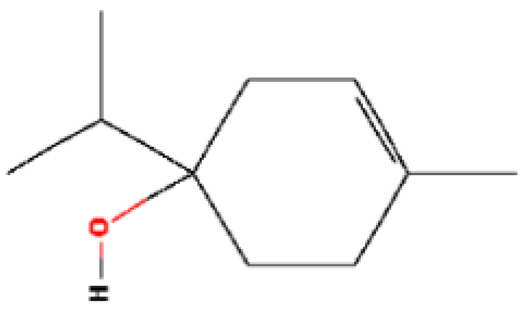
10	α-Terpineol	16.186	17100	98-55-5	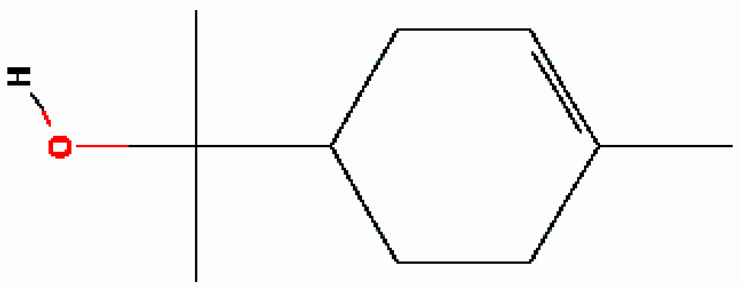
11	(-)-Verbenone	16.519	92874	1196-01-6	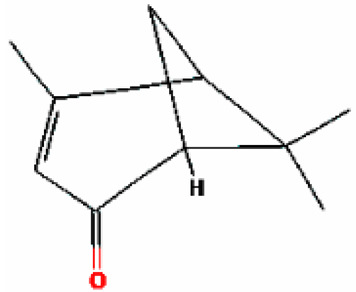

Notes: The naming of all substances is checked against PubChem (https://pubchem.ncbi.nlm.nih.gov/, accessed on 11 March 2021).

**Table 2 ijms-24-03464-t002:** The predicted binding energy (kcal mol^−1^) of DhelOBPs and HIPVs.

Ligands	DhelOBP4	DhelOBP5	DhelOBP6	DhelOBP14	DhelOBP18	DhelOBP20
α-Fenchene	−5.9	−5.4	−6	−6.4	−6.3	−6.5
*p*-Cymene	−6.7	−6.2	−6.1	−7.3	−6.8	−6.5
γ-Terpinene	−6.3	−6.1	−6.3	−7.2	−6.7	−6.6
Terpinolene	−6.5	−6.2	−6.2	−7.4	−6.9	−6.4
Fenchone	−5.9	−6.2	−6.6	−6.7	−6.7	−6.5
Camphor	−5.6	−5.5	−6.3	−6.3	−6.1	−6.8
Pinocamphone	−6.1	−6.2	−6.8	−7.0	−6.5	−6.8
Isopinocamphone	−6.2	−5.7	−6.6	−6.7	−6.5	−6.6
Terpinen-4-ol	−6.2	−6.1	−6.2	−7.1	−6.9	−6.3
α-Terpineol	−6	−6.3	−6.3	−7.1	−6.7	−6.3
(-)-Verbenone	−6.2	−6	−6.5	−6.4	−6.4	−6.7

**Table 3 ijms-24-03464-t003:** IC50 and *K_i_* values of tested HIPVs to DhelOBPs.

Ligands	DhelOBP4				DhelOBP5		DhelOBP6		DhelOBP14		DhelOBP18		DhelOBP20	
	pH 7.4		pH 5.0		pH 7.4									
	IC50	*K_i_*	IC_50_	*K_i_*	IC50	*K_i_*	IC50	*K_i_*	IC50	*K_i_*	IC50	*K_i_*	IC50	*K_i_*
*p*-Cymene	13.0 ± 0.9	12.0 ± 0.9	>50	-	>50	-	ud	ud	>50	-	>50	-	>50	-
γ-Terpinene	11.0 ± 0.3	10.1 ± 0.3	>50	-	>50	-	ud	ud	>50	-	>50	-	>50	-
Terpinolene	15.7 ± 1.8	14.5 ± 1.7	>50	-	>50	-	ud	ud	>50	-	>50	-	>50	-
Fenchone	23.8 ± 5.1	22.0 ± 4.7	47.4 ± 2.6	43.8 ± 2.4	>50	-	ud	ud	>50	-	>50	-	>50	-
Camphor	18.0 ± 2.4	16.6 ± 2.2	>50	-	>50	-	ud	ud	>50	-	>50	-	>50	-
Terpinen-4-ol	27.8 ± 3.5	25.7 ± 3.2	>50	-	>50	-	ud	ud	>50	-	>50	-	>50	-
α-Terpineol	10.4 ± 0.1	9.6 ± 0.1	>50	-	>50	-	ud	ud	>50	-	>50	-	>50	-
(-)-Verbenone	23.1 ± 3.9	21.3 ± 3.6	>50	-	>50	-	ud	ud	>50	-	>50	-	>50	-

Notes: IC50, ligand concentration displacing 50% of the fluorescence intensity of the DhelOBPs/1-NPN complex; *K_i_*, dissociation constant (μM); ‘-’ indicated the binding affinities were too weak (IC50 > 50 μM) to calculate accurate *K_i_* values; ‘ud’ indicated an abnormal fluorescence intensity (increasing) so that *K_i_* values could not be calculated.

**Table 4 ijms-24-03464-t004:** The key residues of DhelOBP4 interacted with HIPVs within 4 Å.

Ligands	Hydrophilic Residues	Hydrophobic Residues
α-Fenchene	-	Leu 11, Leu 32, Met 46, Ile 47, Met 50, Phe 126
*p*-Cymene	Tyr 90	Val 7, Phe 54, Val 56, Ala 68, Phe 71, Leu 72, Met 86
γ-Terpinene	Tyr 90	Val 7, Phe 54, Val 56, Ala 68, Phe 71, Leu 72, Met 86
Terpinolene	His 51	Val 7, Leu 11, Met 50, Phe 54, Phe 71, Phe 126
Fenchone	Tyr 90	Val 56, Ala 68, Phe 71, Leu 72, Met 86, Ile 89
Camphor	His 51, Gly 113	Ile 89, Leu 109, Ala 110, Ile 114, Pro 125, Phe 126
Pinocamphone	-	Leu 11, Leu 32, Ile 47, Met 50, Phe 54, Phe 126
Isopinocamphone	-	Leu 11, Leu 32, Met 46, Ile 47, Met 50, Phe 126
Terpinen-4-ol	Tyr 90	Phe 54, Val 56, Ala 68, Phe 71, Leu 72, Met 86
α-Terpineol	His 51, Tyr 90, Cys 93	Val 56, Ile 89, Ala 94, Leu 109, Ala 110, Phe 126
(-)-Verbenone	-	Leu 11, Leu 32, Met 46, Ile 47, Met 50, Phe 126

## Data Availability

All data included in this study are available upon request by contact with the corresponding author.

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
