# Peer review of "A Highly Expressed Antennae Odorant-Binding Protein Involved in Recognition of Herbivore-Induced Plant Volatiles in Dastarcus helophoroides"

_ijms, 2023, doi:10.3390/ijms24043464_

Round 1

Reviewer 1 Report

Although authors have given some interesting results about the ordors recignation, the novelty of the study is still enough, especically, after checked out some references you cited. The analysis of the underlying physical mechnism for these chemical proceess have yet not given. In addition, some figures and data have been losted in the PDF files, which indicates the attitude of author is not rigorous for this study.

Author Response

Thank you for your suggestions. The figures and data of this paper were revised several times before submission, which may have led to some mistakes. We apologize for this and we have checked the content of the paper again. And we have added the analysis of the underlying physical mechanisms for this chemical process (L328-334).

Reviewer 2 Report

Yi et al. studied the antennae-highly expressed odorant-binding proteins involved in the recognition of herbivore-induced plant volatiles in Dastarcus helophoroides. The work combined GC-MS, behavior analysis, and functional assay in vivo and in vitro to describe the functions of some DhelOBPs highly expressed in the antennae. The strong study evidence provided some positive theoretical support for biological control for the Monochamus alternatus. It needs some minor revision before publication.

1.     Are the compounds of Table 1 and 2-D chemical structures in Fig.1 B the same? If yes, please only number all compounds in Fig.1B (do not show structures there) and list the corresponding chemical structure into Table 1 after each compound.

2.     The layout of Fig.2 and Fig.4 C,D exist some mistakes, please arrange the figures again.

3.     What was the reason of choosing the six OBPs that DhelOBP4, 5, 6, 14, 18, and 20 for the functional studies? According to the binding assay, only DhelOBP4 showed strong binding with candidate ligands.

4.     In Fig. 5, although all six DhelOBPs run the docking analysis with candidate ligands, the following binding assays only showed that DhelOBP4 exhibit a strong binding affinity with ligands, so maybe other DhelOBPs except DhelOBP4 do not need the docking analysis, because making the docking analysis is for the explanation of experimental binding assay.

5.     In Fig. 8, why did each ligand have two figures up and down the chemical name?

6.     Some references’ format is wrong, e.g. Latin name is not in italic.

Author Response

Yi et al. studied the antennae-highly expressed odorant-binding proteins involved in the recognition of herbivore-induced plant volatiles in Dastarcus helophoroides. The work combined GC-MS, behavior analysis, and functional assay in vivo and in vitro to describe the functions of some DhelOBPs highly expressed in the antennae. The strong study evidence provided some positive theoretical support for biological control for the Monochamus alternatus. It needs some minor revision before publication.

  1. Are the compounds of Table 1 and 2-D chemical structures in Fig.1 B the same? If yes, please only number all compounds in Fig.1B (do not show structures there) and list the corresponding chemical structure into Table 1 after each compound.

Reply: Thank you for your advice. The compounds of Table 1 and 2-D chemical structures in Fig.1 B are the same. We have accepted the suggestions and made the changes (Table 1 and Fig.1).

  1. The layout of Fig.2 and Fig.4 C, D exist some mistakes, please arrange the figures again.

Reply: Thank you for your advice. We have made the adjustments (Fig.2 and Fig.4).

  1. What was the reason of choosing the six OBPs that DhelOBP4, 5, 6, 14, 18, and 20 for the functional studies? According to the binding assay, only DhelOBP4 showed strong binding with candidate ligands.

Reply: Thank you for your advice. DhelOBP4, 5, 6, 14, 18 and 20 were selected for functional studies because these proteins have interesting spatiotemporal expression profiles (L168-172). And the results of molecular docking showed that they all have potential binding ability with HIPVs (Table 2).

  1. In Fig. 5, although all six DhelOBPs run the docking analysis with candidate ligands, the following binding assays only showed that DhelOBP4 exhibit a strong binding affinity with ligands, so maybe other DhelOBPs except DhelOBP4 do not need the docking analysis, because making the docking analysis is for the explanation of experimental binding assay.

Reply: Thank you for your advice. As the answer to the previous question. This study is based on the idea of reverse chemical ecology, in which the potential binding ability of target proteins to HIPV is first simulated by docking analysis, and then their binding properties are verified by in vitro binding experiments. Therefore, subsequently, based on their in vitro binding properties, we only analyzed the binding mechanism of DhelOBP4 with HIPVs.

  1. In Fig. 8, why did each ligand have two figures up and down the chemical name?

Reply: Thank you for your advice. This is a typographical problem and we have reworked this image.

  1. Some references’ format is wrong, e.g. Latin name is not in italic.

Reply: Thank you for your advice. We rechecked the format of the references.

Reviewer 3 Report

The paper submitted for review deals with an important topic - olfactory communication among insects. Moreover, the article deals with Dastarus helophoroides. This is an insect species used in biological control of insect pest populations. Hosts of D. helophoroides include Monochamus alternatus, the vector of the parasitic nematode Bursaphelenchus xylophilus. The spread of these nematodes is a global problem, requiring special attention and the search for a variety of methods to reduce the damage they cause.

The article is quite well written. The research was done properly. The conclusions based on the results of the research are appropriate.

The article needs minor corrections before publication.

Comments:

L15: “natural enemies” of what: plants, insects?

L33-36: abbreviated thought. Did the authors mean, the dependence of insect behaviour on environmental signals received through the appropriate organs?

L55: “And they suggested” – who?

L69: change “central median that transmits” to vector

L85: change GS-MS to GC-MS

L89: remove comma after 1

L100: correct the format

L127: insert citation

L133: “were almost expressed in all tissues” – what did the authors have in mind? Correct "tissues" to organ (applies to the entire manuscript).

L143: “adult development” change to “adult physiological state”

L149: change “tissues” to organs. Enlarge the charts. As they are now, they are unreadable on a printout.

L152: enlarge the charts. Figure 4C-D – correct (only part is visible).

L209-214: correct the style. The sentences are not very understandable.

L223: change “noticeable” to significant

L228: explain ns, change p>0.05 to p<0.05

L286: “different developmental stages” - actually, the developmental stages of insects are larva, pupa and imago. Correct (the whole manuscript).

L294-296: style/repetition

L313: “lost their tendency responses” – explain

L317: change ecological to evolutionary

L324: correct “behavioural active substances” - substances cannot behave

L334: as above

L347: explain or correct “woods”

L372: “All dissecting tools have been sterilized with absolute ethanol and RNase-free treatment” – correct the sentence

L384: not “tissue” and not “development”

L420: check grammar

L421: selected on what criteria?

L482: describe the statistical analysis methods used

Supplemental Figure S1 – insert a dot

Supplemental Figure S2: if possible - change the colours of the bars (2 x blue), choose the colours so that the differences are also visible on a black and white printout

Supplemental Figure S3: insert a space after (A)

Supplemental Figure S4: insert a space after (A)

Supplemental Figure S5: insert a space after (A)

Supplemental Figure S6: insert a space after (A)

Supplemental Figure S7: insert a space after (A)

Supplemental Figure S8: insert a space after (A).

Supplemental Table S4: check and standardize font size

Supplemental Table S1 – insert a dot after 1

Supplemental Table S3 – insert a dot after 3

Supplemental Table S4 – insert a dot after 4

Supplemental Table S5 – insert a dot after 5

Author Response

The paper submitted for review deals with an important topic - olfactory communication among insects. Moreover, the article deals with Dastarus helophoroides. This is an insect species used in biological control of insect pest populations. Hosts of D. helophoroides include Monochamus alternatus, the vector of the parasitic nematode Bursaphelenchus xylophilus. The spread of these nematodes is a global problem, requiring special attention and the search for a variety of methods to reduce the damage they cause.

The article is quite well written. The research was done properly. The conclusions based on the results of the research are appropriate.

The article needs minor corrections before publication.

Comments:

L15: “natural enemies” of what: plants, insects?

Reply: Thank you for your advice. We have modified the phrase (L15).

L33-36: abbreviated thought. Did the authors mean, the dependence of insect behaviour on environmental signals received through the appropriate organs?

Reply: Thank you for your question. Yes, this part means that insects rely on the olfactory organs to recognize the signals from the external environment to carry out a variety of physiological activities.

L55: “And they suggested” – who?

Reply: Thank you for your question. This sentence is continuous with the previous sentence, and we have modified the punctuation (L56-57).

L69: change “central median that transmits” to vector

Reply: Thank you for your advice. We have modified the phrase (L72-73).

L85: change GS-MS to GC-MS

Reply: Thank you for your advice. We have modified this abbreviation (L88).

L89: remove comma after 1

Reply: Thank you for your advice. We have modified the punctuation (L92).

L100: correct the format

Reply: Thank you for your advice. We have corrected the format (L109).

L127: insert citation

Reply: Thank you for your advice. We have inserted citation (L136).

L133: “were almost expressed in all tissues” – what did the authors have in mind? Correct "tissues" to organ (applies to the entire manuscript).

Reply: Thank you for your advice. We have corrected "organ" to “organ” in the entire manuscript.

L143: “adult development” change to “adult physiological state”

Reply: Thank you for your advice. We have corrected “adult development” to “adult physiological state” (L153-154).

L149: change “tissues” to organs. Enlarge the charts. As they are now, they are unreadable on a printout.

Reply: Thank you for your advice. We have enlarged the charts.

L152: enlarge the charts. Figure 4C-D – correct (only part is visible).

Reply: Thank you for your advice. We have enlarged the charts (Fig. 4).

L209-214: correct the style. The sentences are not very understandable.

Reply: Thank you for your advice. We have revised (L222-227).

L223: change “noticeable” to significant

Reply: Thank you for your advice. We have changed “noticeable” to “significant” (L236).

L228: explain ns, change p>0.05 to p<0.05

Reply: Thank you for your advice. We have added explanation of “ns” (L241-242) and changed “p>0.05” to “p<0.05” (L242).

L286: “different developmental stages” - actually, the developmental stages of insects are larva, pupa and imago. Correct (the whole manuscript).

Reply: Thank you for your advice. We have corrected “adult development” to “adult physiological state” in the whole manuscript.

L294-296: style/repetition

Reply: Thank you for your advice. We have corrected (L311).

L313: “lost their tendency responses” – explain

Reply: Thank you for your advice. We have added explanations (L328-334).

L317: change ecological to evolutionary

Reply: Thank you for your advice. We have changed “ecological” to “evolutionary” (L337-338).

L324: correct “behavioural active substances” - substances cannot behave

Reply: Thank you for your advice. We have changed “behavioural active” to “attractive” (L345).

L334: as above

Reply: Thank you for your advice. We have changed “behaviourally active” to “attractive” (L356).

L347: explain or correct “woods”

Reply: Thank you for your advice. We have changed “woods” to “wood sections” (L368).

L372: “All dissecting tools have been sterilized with absolute ethanol and RNase-free treatment” – correct the sentence

Reply: Thank you for your advice. We have revised this sentence (L393-394).

L384: not “tissue” and not “development”

Reply: Thank you for your advice. We have changed “Tissue and developmental” to “Spatio-temporal” (L405).

L420: check grammar

Reply: Thank you for your advice. We have revised (L441).

L421: selected on what criteria?

Reply: Thank you for your advice. Based on the principle that the base sequences obtained by sequencing are identical to the transcriptome sequences.

L482: describe the statistical analysis methods used

Reply: Thank you for your advice. We have revised (L487-488).

Supplemental Figure S1 – insert a dot

Reply: Thank you for your advice. We have inserted a dot (Fig. S1).

Supplemental Figure S2: if possible - change the colours of the bars (2 x blue), choose the colours so that the differences are also visible on a black and white printout

Reply: Thank you for your advice. We have change the colours of the bars (Fig. S2).

Supplemental Figure S3: insert a space after (A)

Supplemental Figure S4: insert a space after (A)

Supplemental Figure S5: insert a space after (A)

Supplemental Figure S6: insert a space after (A)

Supplemental Figure S7: insert a space after (A)

Supplemental Figure S8: insert a space after (A).

Reply: Thank you for your advice. We have inserted a space after (A) (Fig. S3-8).

Supplemental Table S4: check and standardize font size

Reply: Thank you for your advice. We have modified the font size (Table S4:).

Supplemental Table S1 – insert a dot after 1

Supplemental Table S3 – insert a dot after 3

Supplemental Table S4 – insert a dot after 4

Supplemental Table S5 – insert a dot after 5

Reply: Thank you for your advice. We have inserted a dot (Table S1-5).